# The DUX-25 after Twenty-Five Years: New Analyses and Reference Data

**DOI:** 10.3390/children9101569

**Published:** 2022-10-17

**Authors:** Hendrik M. Koopman, Benjamin S. D. Telkamp, Annelieke Hijkoop, Julie A. Reuser, Marlous J. Madderom, Hanneke IJsselstijn, Andre B. Rietman

**Affiliations:** 1Department of Clinical, Health and Neuropsychology, Faculty of Social Sciences, Leiden University, 2311 EZ Leiden, The Netherlands; 2Wiskunde & Statistiek Inzichtelijk, 2313 AD Leiden, The Netherlands; 3Pediatric Surgery and Intensive Care, Erasmus MC Sophia Children’s Hospital, P.O. Box 2060, 3000 CB Rotterdam, The Netherlands; 4Child and Adolescent Psychiatry/Psychology, Erasmus MC Sophia Children’s Hospital, P.O. Box 2060, 3000 CB Rotterdam, The Netherlands

**Keywords:** DUX-25, HRQOL, children, chronic illness, proxy, questionnaire

## Abstract

Twenty-five years after its inception, we present new analyses and reference data for the DUX-25, a questionnaire on health-related quality of life for children 8–17 years old and their parents as proxy. Data from 774 healthy children and their caregivers were collected through web-based data collection. Participants were recruited via primary and secondary schools in the Netherlands. The DUX-25 showed adequate psychometric qualities. Using exploratory and confirmatory factor analyses, we were able to support the theorized four-factor model. In addition, a model with five factors emerged in which the factor ‘Social’ was divided into ‘Social Close’ and ‘Social Far’. A comparison of the outcomes of the PedsQL with those of the DUX-25 provides evidence for a high construct validity of the DUX-25. With the new updated reference data, the DUX-25 can still be used in inpatient and outpatient settings to measure health-related quality of life of children with chronic conditions.

## 1. Introduction

Advancements in the medical treatment of children and adolescents suffering from chronic diseases have led to prolonged life expectations for these age groups [1]. The majority of young people suffering from chronic illnesses are now able to lead lives comparable to those of their peers and only need to see a healthcare professional only once or twice a year [2].

Thirty years ago, health-related quality of life (HRQOL) was first shown to be a candidate metric for measuring the effectiveness of health care interventions [3]. Developments since then have made the HRQOL an important outcome measure in the management of chronic diseases [4,5,6] especially in children and adolescents. Health care professionals increasingly started to use HRQOL metrics because of the possibility to identify and prioritize important aspects in the management of chronic diseases. [7].

Nowadays, at least more than 30 disease generic and 60 disease specific HRQOL instruments exist aimed at paediatric populations suffering from chronic illness. One of the less well-known instruments is the DUX-25 [8]. The DUX-25 [9] is a disease-generic, self- and proxy-report instrument, measuring the HRQOL of children suffering from chronic diseases. It assesses the emotional and cognitive evaluation of functioning on four dimensions, namely, physical, emotional, social, and home functioning. Before further improvements led to the publication of the DUX-25, it was known as DUCATQOL [10] and the DUC-25 [11]. Several instruments based on the DUX-25 were developed: A disease-generic instrument for assessing HRQOL in very young children [12], a disease-specific version for young clients with celiac disease [13] and a version for patients suffering from bone tumours [14]. The DUX-25 has also been used as a quality-of-life instrument in various clinical populations in recent years [15,16,17].

The DUX-25 follows the general suggestions for this type of instrument by being brief, easy to administer, and uncomplicated to score [18]. The difference with many other instruments, such as the Pediatric Quality of Life Inventory (PedsQL) [19], is that the DUX-25 measures HRQOL based on the subjective emotional and cognitive evaluation of the impact of impairments in certain areas of life rather than defining how well a child must perform in those areas in order to prove that the child has a high HRQOL. The DUX-25 therefore measures subjective instead of objective well-being [20]. 

Despite the efforts invested in the development of the DUX-25 until now and the limited evidence for its construct validity and internal consistency provided by other authors [21], an original article based on updated reference data from a large sample and an analysis of the instrument’s psychometric properties is still missing.

This article reports on the analysis of psychometric properties of the DUX-25 based on new reference data from 774 children in the Netherlands. Specifically, we evaluated the hypothesis that the DUX-25 is still a reliable (internally consistent and reproducible) and valid (content-, construct- and criterion validity) HRQOL questionnaire.

## 2. Materials and Methods

### 2.1. Participants

The DUX-25 was administered to healthy Dutch children aged 8–17 years and their parent(s)/caregivers(s) between April 2015 and May 2016. Inclusion criteria were: ability to read, understanding of the Dutch language and access to the Internet. Children with a chronic illness or with a Diagnostic Statistical Manual 5 (DSM 5) [22] classification were excluded from analysis. This will allow other researchers to compare a population of children with or without health problems with these norm data.

### 2.2. Measure

The DUX-25 is an HRQOL questionnaire with 25 items divided over four scales, which is completed by either child itself or the parent/caregiver [9]. The parent (proxy) version of the DUX-25 parallels the self-report version for children. The a priori scales are: ‘Body’ (6 items), ‘Emotion’ (7 items), ‘Social’ (7 items), and ‘Home’ (5 items). Responses are scored on a happy-to-sad faces scale using smileys that visualize a 5-point Likert scale (Appendix A). The scores are converted to a 0–100 scale with the sum score being calculated by taking the average of all item scores. A higher sum score represents a better appraisal of HRQOL. The DUX-25 has been validated on the same reference sample as the TACQOL and showed adequate correspondence [9]. The DUX-25 has good overall reliability (*α* = 0.91) and test–retest correlations [10,11].

### 2.3. Procedure

We used a cross-sectional online study design and data were collected through web-based administration. All healthy respondents were recruited through regular primary and secondary schools in the Netherlands, and stratified by location (city or rural) and region (provinces). The selection of schools was stratified in this way and we used data from all healthy respondents without further selection. Parents of children attending the participating schools were sent a letter explaining the purpose and procedure of the study. The letter contained a link to the website of the Erasmus University Medical Center (Erasmus MC) where parents were asked to provide informed consent. Participating parents filled out their email address, and stated the number of children they wished to participate with, and the ages of the children. A researcher (JAR, MJM, or AH) assessed the completed forms and sent both parent and child/children the correct links to the DUX-25 questionnaire (based on age). The parent and child links were provided with the same personal token, which allowed for the anonymous linking of parent and child responses. Children aged 12 years and older were asked to provide informed consent before starting the questionnaire. The layout of the web-based versions of the DUX-25 questionnaires matched the paper versions as closely as possible, except that questions were presented four at a time rather than all at once, and missing values were not accepted.

### 2.4. Socio-Demographic Questions 

The socio-demographic questionnaire (only to be filled out by the parent(s)/caregiver) consisted of three items about the parent(s)/caregiver and four items about the child. Parents were asked for their sex, country of birth, and highest completed education (low, middle or high, classified according to the International Standard Classification of Education (ISCED), 2011) [23]. Items about the child encompassed: the month and year of birth, sex, presence of chronic disease, and presence of psychological problems. 

The study was approved by the Erasmus MC Medical Ethics Committee on 21 April 2015 (MEC-2015-244). 

### 2.5. Statistical Analysis

We hypothesised that the 25 manifest variables (of the DUX-25 questionnaire) are indicators of HRQOL. We expected that the theoretical construct of HRQOL has four underlying dimensions (latent variables). These latent variables represent attitudes related to the constructs ‘Body’, ‘Emotion’, ‘Social’ and ‘Home’. 

The data were checked for structure detection (factorability of data) using Kaiser-Meyer-Olkin Measure of Sampling Adequacy (KMO) and Bartlett’s test of sphericity. All analyses in this study are run with the statistical software program R (version 3.5.0) [24]. 

Parallel Analysis was run with R-Package ‘psyche’ [22] to explore which factor solution is most adequate by using Exploratory Factor Analysis (EFA). Oblique rotations were used when running the EFA models with ‘Maximum Likelihood’ as the estimation method with R-packages ‘psyche’ [25]. The factor solutions were tested with Confirmatory Factor Analysis (CFA) using the R-package ‘Lavaan’ [26]. The difference in fit was tested with a Chi-square test and further evaluated using the goodness and badness of fit measures (Comparative Fit Index (CFI) and Root Mean Square Error of Approximation (RMSEA)). 

To find to what extent the two age groups (8–12 and 13–17 years) within sex (boy and girl) are invariant (the same) on the factor structure (configured), factor loadings (metric), item means (scalar) and item variances (strict), two Multigroup CFA’s were performed. Since the DUX-25 items are measured at an ordinal level (5-point Likert-scales) and since most items show highly left skewed distributions, and CFA assumes the scored items to be at interval (or higher) measurement level with approximately normal distributions, we performed a Polytomous Item Response Theory technique additionally to evaluate and validate the five subscales from the preferred factor solution. 

To further validate the total scale (the average of all 25 items) and its subscales, reliability measures (Cronbach’s Alpha) and correlations with the PedsQL total and subscales and the three additional questions of the DUX-25 were calculated. 

The effects of sex and age group on the subscales (raw subscales and the latent factor scores derived from the factor solution) were examined with a two-way Multivariate Analysis of Variance (MANOVA) followed up with two-way (Univariate) Analysis of Variance (ANOVA).

A multilevel regression was run to compare scores of boys, girls and their fathers or mothers on the DUX-25 total scale and also to investigate (and or correct for) the effects of age and Socioeconomic Status (SES), keeping into account the paired nature of the data (child and his or her mother or father). In this multilevel regression model, the DUX-25 Total Score (of the child itself and the proxy) is at the first level and families are at the second level (in which child and parent are nested). In this analysis, the Total Score as a dependent variable was regressed into the independent variables: factor family member (six categories: boy, his father, his mother, girl, her father and her mother), age (centred), and SES (dummy-coded). For the multilevel regression, the R-package ‘lme4’ [27] was used. Similar analyses were run for the subscales; however, for the sake of simplicity, only age (centred) and family member (three categories: child, father, mother) served as predictors. 

Additionally, to help clinicians compare clients’ scores with population norms, quantile regressions of the DUX-25 Total Score onto age and differentiated for sex were run. 

## 3. Results

The sample consisted of 1361 observations; 593 boys and girls filled in the DUX-25 self-questionnaire and 768 parents completed the proxy-questionnaire. We counted 774 unique families. See Table 1.

### 3.1. Exploratory Factor Analyses

We explored and compared a four, five, and six factor solution through EFA. Based on all children (*n* = 593), the three models were reasonable approximations of the data according to the RMSEA and CFI indices. The 6-factor solution had the lowest and best RMSEA value (RMSEA = 0.040, 90% CI [0.032, 0.045]). The RMSEA values for the four and five factor solutions also indicated a good fit with the RMSEA for the 5-factor solution being the lowest (RMSEA = 0.046, 90% CI [0.039, 0.051]. Although the 6-factor solution had the best fit statistically (CFI = 0.972) [22], this solution contains one factor with only two items (item 1 ‘School’ and item 20 ‘Teachers’) with substantial loadings (>0.30). The CFI’s for the four and five-factor solutions were somewhat lower but both still good to excellent (respectively 0.935 and 0.957). The 4-factor solution explained 43% of the total variance in test scores and the 5- and the 6-factor solution, respectively, explained 45% and 48% of the total variation. Because the 5-factor solution shows an extra distinction on the construct ‘Social’ (‘Social Close’ and ‘Social Far’) that seems clinically relevant and because this solution fits the data statistically better than the 4-factor solution (∆*χ*^2^ (20) = 96.65, *p* < 0.001), we concluded that HRQOL, as measured by the DUX-25 questionnaire, is best represented by five underlying dimensions: ‘Emotion’, ‘Social Close’, ‘Social Far’, ‘Home’ and ‘Body’.

### 3.2. Confirmatory Factor Analyses

In CFA, the 5-factor model showed a significantly better fit than the 4-factor model ($\Delta \chi^{2}$ (4) = 17.76, *p* = 0.001). See Table 2 and Appendix A. Both RMSEA values were sufficient or good with the RMSEA having (rounded) the same value (RMSEA = 0.064). For both models, CFI’s were close to acceptable (CFI _4 factor_ = 0.876 and CFI _5 factor_ = 0.878). As expected, the five factors were strongly associated with each other. The lowest correlation (*r* = 0.49) was found between the factors ‘Social Close’ and ‘Body’ and the highest between ‘Emotion’ and ‘Home’ (*r* = 0.82). All hypothesized relations between indicator variables and latent variables for both models were positive and highly significant with all *p*-values being smaller than 0.001. For the model with five factors, the standardized regression weights ranged between 0.476 (for the relation between item 20 ‘Teachers’ and ‘Social Far’) and 0.865 (for the relation between item 13 ‘Body’ and the factor ‘Body’). 

### 3.3. MANOVA and Corresponding ANOVA’s 

Given that the 5-factor model is an acceptable approximation of the data for all children, we compared the five subscales with a Two Way MANOVA, with Sex, Age Group and their interaction as the independent variables. See Table 3 and Appendix A. Multivariate significant effects were found for Sex and Age Group (*p* < 0.001), and a marginal significance for the interaction was found (*p* = 0.064). 

Univariate follow-up analyses (two-way ANOVA’s) showed that Sex was only associated with significant mean differences for the subscale ‘Body’ (F (5, 589) = 13.18, *p* < 0.001, η^2^_partial_ = 0.02). Boys scored higher on average (*M* = 90.07, *SD* = 18.14) than did girls (*M* = 83.18, *SD* = 21.96). Interestingly, for ‘Body’, also the main effect of Age Group and the interaction was significant. Older children on average scored higher on ‘Body’ than did younger children, but the difference between younger and older children was larger for girls than for boys. For the subscales ‘Emotion’, ‘Social Far’ and ‘Home’, only the main effect of Age Group was significant (all *p*-values < 0.01). No significant effects were found for the subscale ‘Social Close’.

### 3.4. Correlations

For further validation, reliability measures (Cronbach’s *α*) for all DUX-25 scales (Total Scale for child and parent and the 5 subscales for child) and Pearson correlations were calculated between the DUX-25 Scales and the PedsQL Scales. See Table 4. Furthermore, we checked how these measures were related to the three additional questions (child example)—26. How are you?—27. What do you think of your health?—28. What do you think of these questions?

Reliability was acceptable with a range between *α* = 0.59, for ‘Social Close’, and *α* = 0.94 for the Total Score for parents. The DUX-25 Total Score (Self), and the subscale for ‘Emotion’ showed the highest associations with the Total score for the PedsQL (respectively *r* = 0.51, *p* < 0.001 and r = 0.52, *p* < 0.001). The lowest correlation, but still significant, was found between ‘Social Far’ and ‘Body’ (*r* = 0.11, *p* < 0.001). 

### 3.5. Quantile Regression

Additionally, for the DUX-25 Total scale, a quantile regression was run in order to give suggestions for possible cut-off scores as a function of age. See Figure 1. With the data from Figure 1 and Table 5, clinicians are able to compare scores of individual children and parents with these reference data.

Since the presence of heavily negatively skewed distributions and the presence of heteroscedasticity (older children show a wider range of scores), merely regressing the Total Score onto age, which gives the mean Total Score value as a function of age, would not give a useful representation of the data. We examined how the 5th, 10th, 25th (1 Quartile) and the 50th (Median) percentile shift when age increases, separately for boys and girls. For girls, the 10th and the 25th percentiles showed a steeper decrease over age than did the 5th and the 50th percentiles. For boys this also seems to be true, but only for the 25th percentile. As an example, for interpretation of this figure it could be said that about 10 percent of girls of age 16, are expected to have a score lower or equal to about 55 and only 5 percent will have a score lower than (or equal to) approximately 52.

## 4. Plain English Summary

The DUX-25 is a questionnaire on health-related quality of life for children aged 8–17 years. It can be completed by parents or by the children themselves. The answers are scored on a scale from happy to sad faces using smileys. This questionnaire was first published 25 years ago and now new analyses and reference data are presented. Seven hundred seventy-four healthy primary and secondary school children and their caregivers in the Netherlands completed the questionnaire via the internet with the aim to provide norm scores for the DUX-25. 

We present figures and tables to compare the norm scores with the scores of children with a variety of disorders or conditions. Using different types of analyses, a model with four factors was confirmed with scales for ‘Body’, ‘Emotion’, ‘Social’, and ‘Home’. The ‘Social’ scale was subdivided into ‘Social Close’ and ‘Social Far’. The results of a different quality of life questionnaire, the PedsQL, were highly similar to those of the DUX-25. However, the PedsQL focuses on the more objective functioning of children and the DUX-25 focuses on the subjective emotional well-being of the child. In conclusion, the DUX-25 is well suited to monitor children’s well-being and may serve as an outcome to assess the effect of interventions in healthcare settings. 

## 5. Discussion

We reported on the psychometric properties of the DUX-25, based on data from 768 children. Using EFA and CFA, our analyses (supported by IRT analysis) supported the theorized four-factor model. In addition, a model with five factors emerged in which the factor ‘Social’ was divided into ‘Social Close’ and ‘Social Far’. The comparison of the outcomes of the PedsQL with those of the DUX-25 provides evidence for a high construct validity of the DUX-25. In conclusion, this study has demonstrated that the adequate psychometric properties of the DUX-25 render it a reliable instrument for routine outcome monitoring of children’s HRQOL in a clinical setting. 

Reliability of the total scale (both child and proxy versions) proved appropriate for individual interpretation in clinical practice. The subscales can be used to detect areas of concern in a child’s HRQOL, either socially (close or far), emotionally, physically or home-related. To further validate the DUX-25 externally and to develop a more concrete and reliable idea about the implications of a score on the DUX-25, subscales may be used complementary. 

Investigating the factorial structure of the DUX-25, the outcomes yielded by EFA and CFA supported the existence of the originally hypothesized four dimensions [10,11] by loading with the same group of respective items on four factors with good validity. Further analysis of the outcomes pointed us to a five-factor structure where the original ‘Social’ dimension was split up into ‘Social Far and ‘Social Close’ (based on age and frequency of contact).

We considered this additional investigation necessary because EFA and CFA rely on the analysis of interval data, whereas Likert scales provide ordinal data. Conducting parametric analyses on non-parametric data allows using tests that are more sensitive to an effect. However, while some authors claim the soundness of doing so [28], others strongly advise against it, as this approach may produce effects that cannot be statistically substantiated due to the violation of assumptions [29]. We therefore followed the suggestions [30] to leverage the higher sensitivity of parametric tests for a pilot investigation; yet, these suggestions point at the necessity to confirm findings generated this way using a non-parametric test. Furthermore, as the literature in this field features many publications that only provide the outcome of parametric tests, we also publish these outcomes for sake of comparison. 

While EFA, CFA and IRT all support the interpretation of the DUX-25 outcome based on five distinctive factors, the two more detailed social dimensions must be interpreted more carefully than the original single dimension. Splitting up the items of one of four factors from a relatively short instrument leading to the two new social factors, negatively impacts validity. 

Our analyses pointed to two limitations of the DUX-25. With respect to applicability, we recognised that measuring the HRQOL in children under the age of eight with this instrument might lack validity and reliability. The main reason for this, as argued by other authors [31], can be found in the immaturity of cognitive development that makes it challenging for children under the age of eight to fully understand specific concepts of illness and its consequences. However, having only parents assess HRQOL related aspects for these very young children also means that the child’s subjective perspective as essential information to focus interventions on is missing [32]. Therefore, a specific instrument based on the DUX-25 was developed [12] to use for this age group. A second limitation is the ceiling effect of the Total- and the subscale scores of the DUX-25. Since our analyses did not show a floor effect, this means that the DUX-25 has good properties to detect HRQOL changes in the low and mid-range, but presents difficulties in detecting changes when baseline HRQOL is already relatively high. 

Due to the distribution of SES groups in the schools supporting the recruitment process, the SES of most of the candidate respondents was high. Unfortunately, the largest percentage of non-responders per SES group was among those with a low SES, leading to an SES distribution that does not reflect that in Dutch society. Despite huge efforts to encourage all parents to participate, this appears to be a common phenomenon in health-related studies [33]. This structural problem in the research of HRQOL instruments is especially problematic, as clients with lower SES also show lower adherence to more complex health regiments compared to those with higher SES [34,35]. Additionally, health professionals can be biased by the SES when applying disease management decisions [36]. These two aspects indicate that conclusions on HRQOL based on the DUX-25 should be drawn with great caution in clients with a lower SES. 

Another factor potentially limiting the generalizability of our data was found in the distribution of responders per region of the Netherlands. The number of participants living in metropolitan versus rural areas as well as the number of responders per province did not match the distribution of the general population in the Netherlands [37,38]. Citizens living in rural areas, for example in the United States, are considered a ‘health disparity population’ due to lower life expectancy and higher disease rates. Furthermore, access to the healthcare system is more problematic there, as there seem to be more financial problems in rural areas, so more people do not have health insurance and the number of health workers per capita is lower in rural areas. 

Furthermore, only every tenth of the proxy-questionnaires was filled in by a father. Although this seems to be in line with comparable studies, the implications of this common phenomenon are uncertain. While some authors found no difference between fathers and mothers in the evaluation of a child’s QOL [39], others found only moderate agreement between parents with respect to a child’s HRQOL [40]. Others [20] argued that information given by the mother might lead to a broader insight into the HRQOL of children, as they on average observe their sons or daughters in more contexts than do fathers.

In addition to the adequate psychometric properties of the DUX-25 reported above and the original focus being on the subjective evaluation of well-being, the DUX-25 provides several practical advantages. First, the self-explanatory scales and their short length allow for a quick and easy application without supervision, e.g., preceding every appointment. In addition, the total-scores and the sub-scale scores, with the cut-off values as indicated in Figure 1, provide a quick and easy understanding of a child’s HRQOL that needs no further interpretation. It should be noted that the sub-scale ‘Body’ for which scores differed between sexes. Other than this score, most scores produced by the DUX-25 speak for themselves. Our analyses showed that valid conclusions can be drawn from information based solely on proxy questionnaires. This will be relevant for young children and children who suffer from a disease that prevents them from completing the questionnaire themselves.

We examined the construct validity by comparing the DUX-25 by comparison with the PedsQL. This revealed high correlations between the scores for the respective domains produced by both instruments. This is relevant in that the DUX-25 focuses on a client’s emotional and cognitive evaluation to measure HRQOL, while the PedsQL uses functioning as an indicator. The main difference between these concepts is that while functioning attempts to measure well-being by defining specific concepts externally and objectively, the subjective emotional and cognitive evaluation indicates how an individual actually experiences limitations in specific areas of his or her life [41]. In certain clinical groups, the subjective and the more objective view of health can produce different results over time, as we have already substantiated in a study in children after oesophageal atresia repair [20]. When it comes to coping with the consequences of a serious disease, subjective evaluation of certain life domains can lead to important insights. Moreover, a growing body of evidence has linked increased subjective well-being with better health [42], although other authors have also provided contradictory evidence [43].

## 6. Conclusions

The DUX-25 is generally applicable to various age groups and sexes. The adequate psychometric properties, the ease of use in assessment and scoring, and the theoretical basis for measuring well-being based on a subjective evaluation still make the DUX-25 a useful instrument for assessing HRQOL in children 25 years after its inception.

## Figures and Tables

**Figure 1 children-09-01569-f001:**
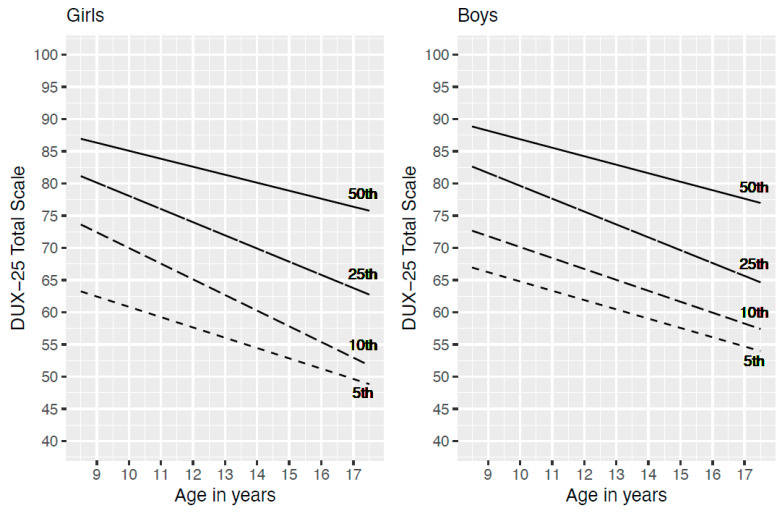
5th, 10th, 25th and 50th Percentile for DUX-25 Total Scale by Age.

**Table 1 children-09-01569-t001:** Demographics.

Group	Frequency	Percentage
Total Number of Unique Families ^a^	774	100.0
**Gender Child**		
Boy	347	44.8
Girl	427	55.2
**Age Group** 8 to 12 years	299	38.6
13 to 17 years	475	61.4
**SES** Low (ISCED 0–2)	64	8.3
Middle (ISCED 3–4)	174	22.5
High (ISCED 5–8)	536	69.3
**Parents Dutch or Foreign** Both parents Dutch	655	84.6
One of the parents Foreign	96	12.4
Both parents Foreign	21	2.7
One parent Dutch other Unknown	2	0.3
**Province of School** Noord-Holland	124	16.0
Zuid-Holland	494	63.8
Other Provinces	156	20.2
**Self** Boy	256	18.8
Girl	337	24.8
**Proxy** Father	106	7.8
Mother	662	48.6
Total Number of Observations	1361	100.0

^a^ Total sample size (N = 1361) consists of 587 child/parent pairs (n = 1174) and 187 single (either child or parent) observations.

**Table 2 children-09-01569-t002:** Standardised Factor loadings for the 4 and 5 Confirmatory Factor Model.

Standardised Factor Loadings (Pattern Coefficients) ^a^										
	4 Factor Model				5 Factor Model					
Item	Emotion	Social	Home	Body	Emotion	Social Close	Social Far	Home	Body	Item Number
**Emotion**										
school	0.605				0.606					1
I often feel	0.754				0.755					2
feel now	0.659				0.659					4
wake up	0.497				0.495					14
things I think	0.673				0.672					17
school work	0.542				0.542					22
at night in bed	0.616				0.616					25
**Social**										
children class		0.561				0.601				5
my friends		0.457				0.526				16
with someone		0.527				0.564				24
adults		0.517					0.535			3
other people		0.616					0.641			8
other children		0.670					0.680			12
teachers		0.462					0.476			20
**Home**										
at home			0.744					0.744		6
father			0.564					0.564		7
My life			0.723					0.724		15
Together home			0.684					0.683		18
mother			0.578					0.578		19
**Body**										
Stamina				0.551					0.552	9
Things I do				0.568					0.568	10
Appearance				0.737					0.736	11
Body				0.865					0.865	13
Height				0.509					0.509	21
weight				0.779					0.779	23

^a^ All factor loadings (standardised partial regession weights) significantly deviate from 0 with *p* < 0.001, N = 593.

**Table 3 children-09-01569-t003:** Multi Level Model, DUX-25 Total Score by Family Member, Age, SES and Country.

Dependent Variable: DUX-25 Total Score							
	Fixed Effects				Random Effects		
		95% Confidence Interval					
Group or Predictor	Estimate ^ab^		Lower Bound	Upper Bound	Group	Name	*SD*
**Groups ^c^**							
Girl	79.58		77.56	81.60			
Her Mother	80.45		78.45	82.45	Betw. Family (sec. level)	Intercept	9.61
Her Father	78.60		74.99	82.20			
Boy	81.58		79.49	83.68	With. Family (first level)	Residual	8.23
His Mother	81.21		79.17	83.26			
His Father	78.95		75.72	82.17			
**Predictors**							
Age (mean centered)	−1.47	***	−1.81	−1.14			
SES Low (ISCED 0-2)	−1.51		−4.83	1.81			
SES High (ISCED 5-8)	−0.76		−2.74	1.21			
Parent Foreign	−0.88		−4.44	2.67			
Other Parent Foreign	0.00		−0.02	0.01			

^a^ Estimates represent ‘means’ for Groups, and ‘slopes’ (*b*) for Predictors. ^b^ *** *p* < 0.001. ^c^ Reference Group: Age = ‘13 years’, SES = ‘SES Medium (ISCED 3–4)’.

**Table 4 children-09-01569-t004:** Bivariate Correlations ^abcd^ between DUX-25 and PedsQL Scales.

	DUX-25 Scales							
		Total Child	Total Parent	Emotion	Social Close	Social Far	Home	Body
	**Cronbach’s** *–α*	0.91	0.94	0.82	0.59	0.67	0.80	0.83
**PedsQL Scales**								
PedsQL Total Child	0.85	0.51	0.36	0.52	0.27	0.28	0.36	0.45
PedsQL Total Parent	0.88	0.37	0.46	0.34	0.23	0.25	0.27	0.31
Body Functioning	0.73	0.30	0.21	0.27	0.12	0.11	0.19	0.35
Emotion Functioning	0.79	0.42	0.28	0.48	0.19	0.19	0.29	0.35
Social Functioning	0.74	0.41	0.28	0.35	0.39	0.28	0.29	0.33
School Functioning	0.68	0.41	0.30	0.46	0.14	0.27	0.30	0.30
**Child**								
26. How are you?		0.73	0.47	0.70	0.47	0.44	0.62	0.56
27. What do you think of your health?		0.53	0.37	0.42	0.33	0.31	0.37	0.57
28. What do you think of these questions?		0.36	0.29	0.31	0.24	0.35	0.27	0.26
**Parent**								
26. How does your child think he or she is doing?		0.46	0.74	0.43	0.31	0.29	0.42	0.34
27. How does your child think of his or her health?		0.40	0.60	0.33	0.21	0.25	0.34	0.36
28. What does your child think of these questions?		0.30	0.46	0.27	0.21	0.24	0.22	0.23

^a^ Pairwise Correlations are based on sample size ranging between 581 and 768. ^b^ All correlations (except one) deviate significantly from 0 with *p* < 0.001. ^c^ The correlation between Body Functioning and Social Far deviates from 0 with *p* = 0.009. ^d^ All scales are for children, unless when mentioned.

**Table 5 children-09-01569-t005:** Descriptives, DUX-25 Scale Means by Family Member and Age Group.

			Total		Emotion		Social Close		Social Far		At Home		Body	
Family Member	Age Group	*n*	*M*	*SD*	*M*	*SD*	*M*	*SD*	*M*	*SD*	*M*	*SD*	*M*	*SD*
**Girls**	**8 to 17 year**	337	**79.52**	12.35	**75.70**	16.24	**86.99**	13.42	**74.57**	13.80	**88.01**	14.32	**76.45**	18.91
	**8 to 12 year**	145	**83.13**	10.08	**78.40**	14.15	**88.68**	12.56	**76.90**	13.79	**91.34**	11.29	**83.19**	14.59
	**13 to 17 year**	192	**76.79**	13.20	**73.66**	17.42	**85.72**	13.92	**72.82**	13.58	**85.49**	15.81	**71.35**	20.20
**Boys**	**8 to 17 year**	256	**80.85**	12.27	**76.70**	14.79	**86.39**	13.78	**73.78**	15.75	**87.48**	13.76	**82.10**	15.80
	**8 to 12 year**	92	**83.13**	11.64	**78.38**	14.11	**86.96**	14.05	**76.43**	14.94	**90.92**	11.46	**84.74**	15.46
	**13 to 17 year**	164	**79.57**	12.46	**75.76**	15.11	**86.08**	13.66	**72.29**	16.05	**85.55**	14.58	**80.61**	15.84
**Mothers**	**8 to 17 year**	662	**80.19**	13.36	**77.78**	15.79	**84.09**	15.32	**76.39**	16.13	**84.55**	15.12	**79.97**	17.54
	**8 to 12 year**	268	**83.76**	11.89	**80.76**	14.54	**86.16**	13.99	**79.90**	15.91	**89.12**	12.12	**84.19**	16.12
	**13 to 17 year**	394	**77.77**	13.77	**75.76**	16.30	**82.68**	16.02	**74.00**	15.86	**81.43**	16.15	**77.10**	17.91
**Fathers**	**8 to 17 year**	106	**77.30**	15.29	**75.20**	17.74	**81.37**	15.83	**73.58**	17.04	**80.47**	16.97	**77.56**	17.44
	**8 to 12 year**	31	**86.58**	11.57	**85.60**	13.57	**89.25**	13.81	**84.07**	14.24	**90.16**	10.29	**85.08**	15.80
	**13 to 17 year**	75	**73.47**	15.05	**70.90**	17.54	**78.11**	15.55	**69.25**	16.27	**76.47**	17.61	**74.44**	17.23

## Data Availability

Anonymized data are available upon reasonable request by contacting the corresponding author.

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
