# Peer review of "The DUX-25 after Twenty-Five Years: New Analyses and Reference Data"

_children, 2022, doi:10.3390/children9101569_

Round 1

Reviewer 1 Report

The proposed work present new analyses and reference data for the DUX-25, a questionnaire on health-related quality of life for children 8-17 years old and their parents as proxy, twenty-five years after its inception by same author Hendrik M. Koopman (2001).

The sample size is valid (N = 774), covering healthy children and their caregivers were collected through web-based data collection. Participants were recruited via primary and secondary schools in the Netherlands.

It allows us to obtain relevant data to know the CVRS de los niños sanos and their emotional and psychological affectation, a very relevant factor in the pathology and even on many occasions during the same, constituting in itself an important public health problem. These data also highlight the need to promote health and health policies that include mental psychosocial factors as part of the assessment and treatment of these patients and, in general, of chronic pathologies that affect quality of life, and also serve to invest in research in this field.

The statistical techniques used are appropriate for the purpose of the study. Results highlight the need for this type of study, that contribute to decision-making in the management of the disease and promote health policies to improve the quality of life of people and serves to invest in research in this field.

In the following, I make some remarks on theoretical and methodological aspects that may be useful to the authors:

Literature review:

- The literature review, which is shown is adequate and current, situates the reader in the background of the researched topic, although I think it could be completed with  some more updated studies in 2019-2022 and some referent studies of populations in other countries, allowing:

 to establish comparisons with other countries that have also studied the same area, in order to give the widest coverage to the population under study.

Methods and results:

Methods

It is necessary to specify the design of the study and the type of sampling used

It is also advisable to specify the criteria for inclusion (Could children's cognitive ability be a criterion for inclusion?.)

I have reservations as to whether the objective of the study. If the DUX 25 instrument measures health-related quality of life in children with chronic pathologies, why were healthy children selected and why was having a pathology considered as an exclusion criterion?

I would be very grateful if the authors could clarify this question?

Could the study of the Dux-25 questionnaire be complemented by another measure of health-related quality of life?

Results

The results are very clearly expressed. The statistical techniques are adequate both for the control of variables, as well as for the treatment of the data obtained and the robustness of the data.

Conclusions:

The study has all the conditions required in research and provides very interesting data that will help to improve the quality of life of the people under study.

It should also be noted that this type of survey provides us with general data on certain dimensions related to the health of the population but does not always adequately specify each dimension.

I encourage the authors to continue their research along these lines.

Author Response

Dear reviewer #1

We are grateful for your reviewing our paper and for your kind comments. We hope we have addressed all your points of concern in the paper itself and in our answers to your comments below. We also believe that integrating this information improved this report of our study.

Literature review:

  • The literature review, which is shown is adequate and current, situates the reader in the background of the researched topic, although I think it could be completed with  some more updated studies in 2019-2022 and some referent studies of populations in other countries, allowing to establish comparisons with other countries that have also studied the same area, in order to give the widest coverage to the population under study.

Response: Thank you for this suggestion. We added several studies using the DUX-25 in recent years, both from studies in our country and from other, neighbouring countries in lines 52-54.

Methods

  • It is necessary to specify the design of the study and the type of sampling used

Response: We added design and sampling method to the Procedure-chapter, lines 88 and 91-92.

  • It is also advisable to specify the criteria for inclusion (Could children's cognitive ability be a criterion for inclusion?)

Response: Since we only used data from regular (not special) schools (as added to line 89), we do not expect to have included any children with intellectual disabilities. Further inclusion criteria were specified in lines 71-75.

  • I have reservations as to whether the objective of the study. If the DUX 25 instrument measures health-related quality of life in children with chronic pathologies, why were healthy children selected and why was having a pathology considered as an exclusion criterion? would be very grateful if the authors could clarify this question?

Response: The aim of this study is to collect Dutch reference data for the DUX-25. This will allow other researchers to compare a population of children with or without health problems with these norm data. We added this information in lines 75-76.

  • Could the study of the Dux-25 questionnaire be complemented by another measure of health-related quality of life?

Response: As we stated in lines 56-62, we believe the DUX-25 measures quality of life and not health status, which are different but indeed complementary concepts. To substantiate and validate this, we compared our results with those of the PedsQL in this paper (lines 214-225) and in other studies (lines 339-342).

Reviewer 2 Report

The article is very interesting, it confirms that DUX-25 is a useful tool for the assessment of HRQOL in children 25 years after its creation.

It's well organized study.

I have very few comments - rather technical:

Line 106: No date of approval of the bioethical commission

Line 395-482: References to be improved in line with guidelines.

Author Response

Dear reviewer #1

We are grateful for your reviewing of our paper and for your kind comments. We hope we have addressed all your concerns in the paper itself and in our answers to your comments below.

  • Line 106: No date of approval of the bioethical commission

 Response: We added this to lines 113-114.

  • Line 395-482: References to be improved in line with guidelines.

Response: Thank you, we have indeed noticed some imperfections and we hope we have improved the current list of references sufficiently.
